# Interactions between *Magnaporthiopsis maydis* and *Macrophomina phaseolina*, the Causes of Wilt Diseases in Maize and Cotton

**DOI:** 10.3390/microorganisms8020249

**Published:** 2020-02-13

**Authors:** Ofir Degani, Shlomit Dor, Dekel Abraham, Roni Cohen

**Affiliations:** 1Department of Plant Sciences, Migal—Galilee Research Institute, Tarshish 2, Kiryat Shmona 11016, Israel; dorshlomit@gmail.com (S.D.); dekel_abr@hotmail.com (D.A.); 2Faculty Sciences, Tel-Hai College, Upper Galilee, Tel-Hai 12210, Israel; 3Department of Plant Pathology and Weed Research, Institute of Crop Protection, A.R.O., The Volcani Center, Newe Ya’ar, Ramat Yishai 30095, Israel; ronico@volcani.agri.gov.il

**Keywords:** *Cephalosporium maydis*, cotton, crop protection, fungus, *Harpophora maydis*, late wilt, maize, *Macrophomina phaseolina*, *Magnaporthiopsis maydis*, real-time PCR

## Abstract

Fungal pathogens are a significant threat to crops worldwide. The soil fungus, *Magnaporthiopsis maydis*, severely affects sensitive maize hybrids by causing the rapid wilting of plants at the maturity stage. Similarly, the soil fungus, *Macrophomina phaseolina*, develops in a variety of host plants, which leads to rot and plant mortality. The presence of both pathogens together in diseased cotton plants in Israel suggests possible interactions between them. Here, these relationships were tested in a series of experiments accompanied by real-time PCR tracking in maize and cotton. Despite the fact that neither of the pathogens was superior in a growth plate confrontation assay, their co-inoculum had a significant influence under field conditions. In maize sprouts and fully matured plants, infection by both pathogens (compared to inoculation with each of them alone) led to lesser amounts of *M. maydis* DNA but to increased amounts of *M. phaseolina* DNA levels. These results were obtained under a restricted water regime, while optimal water irrigation led to less pronounced differences. In water-stressed cotton sprouts, infection with both pathogens led to an increase in DNA amounts of each of the pathogens. Whereas the *M. maydis* DNA levels in the double infection remain high at the end of the season, a reduction in the amount of *M. phaseolina* DNA was observed. The double infection caused an increase in growth parameters in maize and cotton and decreased levels of dehydration in maize plants accompanied by an increase in yield production. Dehydration symptoms were minor in cotton under an optimal water supply. However, under a restricted water regime, the double infection abolished the harmful effect of *M. phaseolina* on the plants’ development and yield. These findings are the first report of interactions between these two pathogens in maize and cotton, and they encourage expanding the study to additional plant hosts and examining the potential involvement of other pathogens.

## 1. Introduction

The fungus *Magnaporthiopsis maydis* is known to severely affect sensitive maize (*Zea mays*) plants at the maturity stage. It has two additional synonyms, *Cephalosporium maydis* [1] and *Harpophora maydis* [2]. The disease, commonly known as “late wilt”, is reported so far in about eight countries but is considered the most harmful maize disease in Egypt [3] and Israel [4,5]. Although some prevention methods can restrict disease outbreak (for example, see [6]), the use of resistance maize cultivars is still the most common method for minimizing yield losses [7], and it was shown that plant hormones might be involved in restricting the pathogen [8]. Until lately, the only familiar and confirmed alternative host for *M. maydis* was *Lupinus termis* (lupine), which is widely cultivated in Egypt [9]. *M. maydis* was recently identified in *Gossypium hirsutum* (Pima cotton, Goliath cv.), *Citrullus lanatus* (watermelon, Malali cv.), and *Setaria viridis* (green foxtail) [10].

Late wilt is often associated with contamination by secondary plant pathogenic fungi, enhancing the stem symptoms. Indeed, fungi such as *Fusarium verticillioides* causing stalk rot, *Macrophomina phaseolina* causing charcoal rot, and *M. maydis* are grouped in a post-flowering stalk rot complex, which was identified as being one of the most widespread and destructive groups of diseases in maize [11]. The interactions between *M. maydis* and *Fusarium oxysporum* in the roots of cotton and maize were previously investigated [12]. These interactions are associated with an appreciable decrease in the severity of the cotton wilt disease but do not affect the pathogenicity of *M. maydis* to maize. The reduction in infection in cotton was more pronounced when the *M. maydis* precedes *F. oxysporum* in the soil than when they were inoculated simultaneously. *M. maydis* exerts little or no such effect when it follows *F. oxysporum* in the soil. These interactions are particularly interesting since both pathogens can be found abundantly in Egyptian soils and share these two summer crop hosts (cotton and maize), which are cultivated alternately in two-year rotations [12]. Interestingly, both *F. oxysporum f.* sp. lupini and *M. maydis* are also common fungal pathogens in lupine plants in Egypt, causing wilt diseases that result in substantial economic losses. The presence of both pathogens, *M. maydis* and *F. oxysporum*, in diseased maize, cotton, and lupine plants [12] evoke the question of how their interactions or cross-influence affect these and other potential host plants. 

Despite the pivotal role of *M. maydis*, accumulating lines of evidence suggest that it does not act alone; instead, it is part of a more massive complex of maize pathogenic fungi [11]. Some of these fungi, such as *F. verticillioides*, are a secondary invader or opportunist that developed in late wilt-diseased attenuated maize plants [5]. Hence, the co-influence between *M. maydis* and other phytopathogenic fungi has only now begun to be revealed. The current study is an additional step towards uncovering the intriguing relationship between the late wilt pathogen and other phytoparasitic fungi. In particular, the interactions between *M. maydis* and *M. phaseolina* in maize and cotton plants were investigated. 

*M. phaseolina* (Tassi) Goidanich, the charcoal rot causal agent, is a common pathogen affecting a wide range of cultivated and wild species in temperate, warm and tropical regions around the world [13]. The fungus has a vast host range and is responsible for causing damage to more than 500 cultivated and wild plants [14], including maize and cotton. Similar to *M. maydis*, *M. phaseolina* is a primarily soil-borne and seed-borne pathogen, but unlike *M. maydis*, it has a highly competitive saprophytic ability [15]. 

Both fungi have a similar pathogenic mode of action and produce similar visible symptoms. These pathogens are a challenge to control because they often survive in the soil for long periods. When a susceptible host plant is seeded, the fungi can penetrate the plants’ roots, causing root necrosis and affecting sprout development [14,16]. Usually, the symptoms develop later in the season as plants begin to flower and are enhanced under drought conditions [17,18]. When the growth season advances, *M. maydis* and *M. phaseolina* may spread upwards inside the plant’s vascular system, disrupt the water supply, and lead to dehydration and yield reduction [19,20]. While *M. phaseolina* is capable of producing phytotoxins [14], so far, no secreted phytotoxins have been identified in *M. maydis*. However, similar to *M. phaseolina*, *M. maydis* culture filtrate negatively affects maize seed development [21]. 

In a survey conducted in 2016 (Roni Cohen, unpublished results), *M. phaseolina* was identified at a high frequency in cotton fields scattered throughout the Hula Valley in the Upper Galilee (northern Israel). These fields are also known to be heavily infested with *M. maydis*, which causes a severe disease to sensitive maize hybrids. In the summer of 2017, both *M. maydis* and *M. phaseolina* were identified using quantitative real-time polymerase chain reaction (qPCR) in diseased cotton plants in a commercial field in Yavne (southern coastal plain of Israel) [10]. These accumulating lines of evidence suggested possible co-interactions between the two pathogens in cotton and maize in commercial fields where they coexist. 

To the best of our knowledge, this is the first study of the co-influence of *M. maydis* and *M. phaseolina* under field conditions with molecular DNA tracking of each pathogen in maize and cotton tissues. This study aims at deepening our understanding of these curious relationships by measuring changes in the plants’ phenological development, health condition, and yield production. Previous reports of antagonisms among plant pathogens are relatively sparse [22]. The scarcity of reports may belie their frequent occurrence. A recent meta-analysis suggests that these interactions are probably much more prevalent than previously estimated [23]. Reports on pathogen–pathogen antagonisms encompass diverse plant disease systems, including tree pathogens and wood-rotting fungi [24], storage-rotting fungi [25], foliar pathogens [26], and root rots [27]. Studies of foliar pathogens are most common, with antagonisms being reported among foliar pathogens of wheat, barley, and peas (see, for example, [26,28,29]). This may be due to their relative ease of observation and not necessarily due to some innate association. 

## 2. Materials and Methods 

### 2.1. Fungal Isolates and Growth Conditions

The *M. maydis* isolate called *Hm-2* (CBS 133165, CBS-KNAW Fungal Biodiversity Center, Utrecht, the Netherlands) was recovered from wilting maize plants sampled in Sde Nehemia (the Hula Valley, Upper Galilee, northern Israel) in 2001, and identified using pathogenicity, physiology, colony morphology, and microscopic and molecular traits [5,30]. The *M. phaseolina* isolate called *Mp-1* was recovered from wilted cotton plants in 2017 (Roni Cohen’s lab, Newe Ya’ar Research Center, northern Israel) and was identified using pathogenicity, physiology, colony morphology, and microscopic characteristics. Final molecular identification of this isolate was accomplished in this study by qPCR with a primer set targeting *M. phaseolina* species-specific fragments [31], and a primer set targeting the Internal transcribed spacers (ITS), ITS1 and ITS4, as we will elaborate below. The qPCR resultant oligonucleotide was identified by sequencing. The fungi were grown on rich potato dextrose agar (PDA) (Difco, Detroit, MI, USA) at 28 ± 1 °C in the dark for 4-7 days.

### 2.2. Maize and Cotton Cultivars Selected for This Study

The Prelude cv. sweet maize from SRS Snowy River seeds, Australia (supplied by Green 2000 Ltd., Israel) was chosen for this study. The Prelude cv. had been previously tested for susceptibility to late wilt in the field [6,20] and proved to be highly sensitive. The Pima cotton, Goliath cv. (extra-long-staple [ELS] cotton) is commonly grown in different parts of Israel (supplied by Isreal Seeds, Kibbutz Shefaim). This cotton cultivar is also traditionally grown on late wilt contaminated fields during crop rotation and was recently reported to be *M. maydis* vulnerable [10]. In an extensive survey conducted by Roni Cohen across Israel (Newe Ya’ar Research Center, northern Israel), the Pima cotton, Goliath cv. was reported to be sensitive to *M. phaseolina* charcoal rot disease [32].

### 2.3. Plate Confrontation Assay

The interactions between *M. maydis* and *M. phaseolina* when both fungi are grown together on the surface of potato dextrose agar can be referred to as antagonism or a mycoparasitism. The plate confrontation assays were performed by positioning a mycelia disk (6 mm in diameter, taken from the margins of a 4-6-day-old colony) of *M. maydis* at one pole of the culture plate and mycelia disk (at the same size and age) of *M. phaseolina* on the opposite pole. The *M. phaseolina* was added to the plate two days after *M. maydis* since it grows significantly faster. The two fungi were then allowed to grow under optimal conditions (28 ± 1 °C in the dark) for six days until the colonies’ margins met in the area of interaction.

### 2.4. Full-Growth Season Pot Experiments under Field Conditions 

This study examined the combined effect of *M. maydis* and *M. phaseolina* on the growth and yield of maize and cotton plants grown in pots in an open-air enclosure under field conditions. The experiments aimed at simulating field conditions and the reason for using pots (positioned in the open-air field) with naturally infested soil instead of sowing the plants directly to the field soil was to allow enhancing the soil inocula in order to achieve, as much as possible, high and equable infection, and for better control of the water regime. To elaborate on this, pathogenicity trials cannot rely on natural soil infestation alone, which can lead to highly variable results. Even in heavily infested fields, the spreading of the pathogen is not uniform. The pathogen is scattered in small quantities in the soil, and the disease spreading is not uniform in the field.

The experiments were conducted in an experimental farm located near Kibbutz Amir (in the Hula Valley Upper Galilee, northern Israel) during a whole growing season and were subsequently repeated twice in the spring and summer of 2018 and 2019. The two subsequent experiments were performed in a completely randomized design. Each treatment included 10 independent replications (pots). Each pot was 10 L in volume. The negative control in the experiments was soil taken from a nearby field that had no history of *M. maydis* or *M. phaseolina* infestation, and if such an infestation did exist, it was assumed to be very low. All the plants received fertilization and insecticides according to the recommended growth protocol of the Israel Ministry of Agriculture Consultation Service (SAHAM). Each of the pots was seeded with five seeds. The sprouts were diluted to one plant per pot during the seedling growth stage (34 and 23 days after sowing, DAS, in the maize pots, and 28 and 41 DAS in the cotton pots in 2018 and 2019, respectively). Watering was done by drip line irrigation (two droppers per pot) and controlled by a computerized irrigation system. The specific irrigation amount varied for each of the two experiments, as detailed in Table 1. The 2019 repetition included additional treatments to evaluate the effect of the deficient water regime, as will be detailed below. The average meteorological parameters measured during the 2018 and 2019 experimental periods were similar, with higher soil temperature (in both the maize and cotton growth periods) and radiation (only during the maize growth period) in 2019, as detailed in Table 2.

The methodology used for plant inoculation and growth was similar to that of Degani et al., 2019 [20]. The inoculum method involved mixing naturally infested peat soil taken from the Neot Mordechai maize field (Hula Valley, Upper Galilee, northern Israel), which was known to be *M. maydis* infested for many years [33] and was probably also subjected to *M. phaseolina* charcoal rot disease [32], with 30% Perlite No. 4 (to aerate the soil). 

Additionally, a complementary inoculation with the *Hm-2* isolate was carried out in two steps. First, 40 g of sterilized infected wheat seeds were added to the top 20 cm of the soil of each pot with the sowing. These seeds were previously incubated for three weeks at 28 °C in the dark with *M. maydis* or *M. phaseolina* culture agar disks (10 disks per 100 g seeds) and were used here to disperse the pathogen in the soil, as previously described [5,34]. Second, with the above-ground appearance, two agar disks (6-mm-diameter) taken from five-day-old *M. maydis* or *M. phaseolina* colonies (grown for six days at 28 °C in the dark) were added to the upper parts of the roots (4 cm beneath the ground surface). In the maize pots, this procedure was performed 13 or 9 days after the sowing (in the 2018 and 2019 experiments, respectively). In the cotton pots, this procedure was performed 22 or 9 days after the sowing (in the 2018 and 2019 experiments, respectively). In the 2019 repetition, an *M. phaseolina* complementary inoculum with *Mp-1* isolate spore suspension was added to enhance the disease severity outcome. The *M. phaseolina* spore suspension was prepared by washing and collecting the spores from 10 five-day-old *M. phaseolina* colonies grown, as described above, on PDA plates. The spores were then suspended in 1 L sterile double distilled water (DDW). Ten ml of this suspension was sprinkled onto each seed with the sowing.

Maize developmental stages are stated according to [35]. Emergence percentages evaluation conducted 9 DAS for all the plants. The growth parameters were evaluated at the maize sprouting phase (29, 34-37, 54 DAS). Later, at the end of the experiment, the maize harvest day (79-82 DAS), phenological stage evaluation, wilt determination, and yield assessment were performed. A dehydration assessment was done on harvest day by calculating the percentage of plants showing typical maize late wilt dehydration symptoms—the upper leaves’ color alternation to light-silver and then to light-brown and rolling inward from the edges of the leaf. This assessment was done using four categories: 0—the plant is completely dried, 1—the plant has severe dehydration symptoms (over 50% of its part are dehydrated), 2—the plant shows light symptoms and most of its parts are green, 4—the plant is healthy, green, and without visible signs of disease. Thus, each repeat (individual plant) was evaluated according to this scale, and the dehydration proportions are the mean result received from each treatment. A similar symptom evaluation procedure was conducted for the cotton plants 37-41 and 54 DAS, as well as on the harvest day (154 or 167 DAS in 2018 and 2019, respectively), including growth parameter assessment and yield determination. Additionally, all of the plants in each treatment were uprooted, and each plant’s above-ground parts were measured to determine their height and wet weight. The roots’ and shoots’ dry biomass was determined after drying the plants at 65 °C for 62 h. Samples of tissue were taken in order to identify the fungal DNA inside the host tissues using qPCR, as will be described below.

*The 2018 experiment.* In the first experiment (2018), the sowing of the cotton plants was performed on May 5, 2018, and the sowing of maize was performed on May 24, 2018. Both cotton and maize plants emerged above the ground surface eight days after sowing. The maize fertilization took place 57 DAS (about one week after the male flowering). At that age, most of the plants were at the R1 development stage (silking: plants are at a maximum or near-maximum height and have one or more silks extending outside the husk leaves). The maize plants were harvested on August 14, 2018 (82 DAS, 25 days after fertilization, DAF). The cotton plants flowered on May 30, 2018 (58 DAS) and were harvested on October 3, 2018 (154 DAS).

*The 2019 experiment.* The field experiment, described above, was repeated in the summer of the following year (2019) at the same location and according to the same design. This repetition included additional treatments to evaluate the effect of water regime on pathogenesis outcome and disease symptoms development. Thus, half of the treatments received reduced irrigation (see Table 1) in order to produce a restricted, near-drought water regime that may encourage the development of *M. maydis* and *M. phaseolina*, and to facilitate the disease symptoms outbreak. 

The sowing of the maize and cotton plants was performed on May 21, 2019. The maize plants’ fertilization took place 58 DAS (about one week after the male flowering). At that age, in the maize plot that received a regular water supply, most of the plants were at the R1 development stage. At the same age (58 DAS), the maize plots that had received reduced irrigation were at the VT development stage (tasseling: plants with fully visible tassel branches). The low water supply did not allow the plant to continue developing beyond this age (58 DAS), and they suffered from severe drought. Consequently, the reduced water supply plots were harvested one week later, 65 DAS, before the development of cobs. The maize plants that received a regular water regime were harvested on August 08, 2019 (79 DAS, 21 DAF). In the cotton plots, on 58 DAS, the plants had blossom buttons but had not yet flowered. The cotton plants flowered about one week later and were harvested on October 29, 2019 (161 DAS(. 

### 2.5. Molecular Diagnosis

A molecular diagnosis was performed on plants collected from the pots experiment. Roots were washed to remove soil under tap water. The root tissue samples were taken from the uppermost part of the root—from the crown, seminal, and lateral roots (the fibrous root system)—and not from the primary root (the radicle root). The stem tissue samples were taken from the joint section between the aboveground first and second internode. Sampling was done by removing a cross-section of approximately 2 cm in length from each plant stem. The total weight of the samples (roots or stem) was adjusted to 0.7 g and considered to be one repeat. Tissue samples were placed in universal extraction bags (Bioreba, Reinach, Switzerland) with 4 mL of cetyl hexadecyltrimethylammonium bromide (CTAB) buffer and were ground with a manual tissue homogenizer (Bioreba, Reinach, Switzerland) for 5 min until the tissues were completely homogenized. The homogenized samples were treated for DNA purification as previously described [36]. The DNA was suspended in 100 µL of HPLC-grade water and kept at −20 °C until use in the qPCR analysis.

All of the qPCR reactions were performed as previously described [36] using the ABI PRISM^®^ 7900 HT Sequence Detection System (Applied Biosystems, Foster City, CA, USA) for 384-well plates. The 5 µL of total reaction was used per sample well, including 2 µL of sample DNA extract, 2.5 µL of iTaq™ Universal SYBR^®^ Green Supermix (Bio-Rad Laboratories Ltd., Rishon Le Zion, Israel), 0.25 µL of forward primer and 0.25 µL of reverse primer (10 µM from each primer to a well). The qPCR program was as follows: precycle activation stage, 1 min at 95 °C; 40 cycles of denaturation (15 s at 95 °C) and annealing and extension (30 s at 60 °C), followed by a melting curve analysis. Plant samples (root and stem tissues) from each experiment were analyzed separately by qPCR. Each sample was tested three times by qPCR to ensure consistency of the results. The A200a primers were utilized for qPCR (sequences in Table 3). The gene encoding for the last enzyme in the respiratory electron transport chain of the eukaryotic mitochondria-cytochrome c oxidase (*COXI* gene) was used as a “housekeeping” reference gene to normalize the amount of DNA [37]. The *COX* gene was amplified using the COX F/R primer set (Table 3). The amplified gene is from both the plants and the fungi that inoculate it. Relative quantification of the target *M. maydis* fungal DNA was calculated according to the ΔCt model [38]. The efficiency was presumed to be the same for all samples.

### 2.6. Statistical Analyses

A fully randomized statistical design was used in all experiments. Statistics and data analysis were carried out using the JMP program, 7th Edition, SAS Institute Inc., Cary, NC, USA. For the evaluation of *M. maydis* and *M. phaseolina* infection outcome in the experiments, the one-way ANOVA followed by post-hoc multiple comparisons of Student’s *t*-test for each pair was used. The *t*-test compared (with a significance threshold of *p* = 0.05) each treatment to the control.

## 3. Results

### 3.1. Plate Confrontation assay

The relationship between the two soil pathogens that cause maize and cotton wilt diseases was tested in a confront antagonism assay in PDA culture plates. In this assay, a dominant fungus will usually continue to grow behind the confronted line while covering the other fungi. The results of this assay were compared to the growth rate of each fungus alone under the same conditions in order to estimate the growth inhibition rate caused by the other fungus. The growth of *M. phaseolina* was found to be faster than *M. maydis*, so the former was sown onto the plate two days after *M. maydis*. During the six days of incubation in the dark, the fungi grew toward each other until the meeting point, where they formed a clear line between the colonies and no further progress was possible. No mycoparasitism (growth of one fungus on top of the other) was observed (Figure 1A). Moreover, no inhibitory effect by secretion products of one of the pathogens that arrested the other was detected. If such an inhibitory effect does exist, it was effective only at a very short distance (approximately 2 mm) since the two fungi colonies formed a clear line between them (Figure 1B). Additionally, no interlocking of the webs between the two fungi was found.

### 3.2. Full-Growth Season Pot Experiments under Field Conditions

The combined effect of the two soil pathogens, *M. maydis* and *M. phaseolina*, at the late plant’s phenological stages (in which the signs of the disease are most noticeable), was tested in two years of subsequent experiments in maize and cotton. The experiments were conducted in the spring and summer of 2018 and 2019 in pots under field conditions throughout a full growing season. Similar results and tendencies were obtained in both repetitions, but, as expected, some variations occurred in the field conditions, expressed mainly in the intensity of the differences between treatments. Thus, in order to achieve a full and accurate understanding of the interactions between the two fungi, it is important to present the major findings from both years. Table 4, Table 5, Table 6 and Table 7 present the 2018 experiment, while most figures present the 2019 experiment (as will be detailed in each figure legend). In the molecular DNA tracking, the high level of variations within the results due to changes in field environment conditions, together with an objective difficulty to achieve a uniform infection, resulted in relatively high standard error values. Consequently, no statistically significant difference can be measured in most of those tests in comparison to the control. The negative control in the full-growth experiments was soil taken from a nearby field that has no record of *M. maydis* or *M. phaseolina* infestation, and if such an infestation did exist, it was assumed to be minor. Indeed, as will be illustrated in the results, the control field soil contained small quantities of the pathogens.

### 3.3. The Co-Influence of M. maydis and M. phaseolina on Maize Plants

Molecular tracking of both pathogens’ DNA in the maize roots’ tissues during the growing season revealed the effect of the double infection on the spreading of both pathogens at the sprout stage (37 DAS, Figure 2). Infecting the plants with both pathogens, in comparison to infection with each of the pathogens alone, led to a decrease in the presence of *M. maydis* pathogen DNA and an increase in the amount of *M. phaseolina* in the plant roots. The difference in *M. maydis* DNA levels was maximized under water deficiency stress (to 16 times lower levels in the double infection treatment compared to the *M. maydis* inocula treatment, Figure 2 upper panel). In comparison, the highest *M. phaseolina* DNA was reached in the double infection treatment in the drought condition treatment, 3.6-fold higher than the infection solely with *M. phaseolina*. Despite these considerable changes in both pathogens, no measurable influence was reflected in the plants’ growth parameters (34 DAS), which were very similar in all treatments and the control (Table 4). 

Tracing the fungal DNA in maize plant tissues at the end of the growing season (79-82 DAS) revealed a similar picture to that found in sprouts. At the end of the growing season, the combined infection with both pathogens compared to infection with *M. maydis* or *M. phaseolina* alone resulted in a sharp decrease in the fungal DNA levels of *M. maydis* and signaled an elevation in *M. phaseolina* DNA within the roots and the above-ground plants’ tissues (Figure 3). The double infection led to undetectable levels of *M. maydis* DNA in the roots and to 321 times lower levels in the stalks. The shoot’s *M. maydis* DNA level was 10 times higher than in the *M. phaseolina* inoculation, which was similar to the control (a natural field soil with very low levels of infestation, Figure 3). Remarkably, the double infection with both pathogens decreased *M. maydis* DNA levels nearly 20-fold below the control levels. 

Interestingly, the appearance of *M. maydis* DNA in plants that grew on naturally infested field soil enriched with *M. phaseolina* inocula was high in the roots, compared to the *M. maydis* inocula treatment. This is probably the consequence of differences in the presence of *M. maydis* in this soil and natural variability in the sensitivity of the host plants. Indeed, half of the repeats in this treatment resulted in zero levels of *M. maydis* DNA. 

*M. phaseolina* DNA levels in the double infection treatment (compared to sole *M. phaseolina* infection) increased 4.7- and 24-fold in the maize roots and stalks, respectively. It was observed that the overall DNA levels of *M. phaseolina* increased considerably in the roots towards the end of the season and that they were remarkably higher in the roots compared to the stalks (Figure 3).

In the 2018 experiment at this plant age (82 DAS), the fungal DNA variations had a noticeable effect on the plants’ growth parameters and yield (Table 5). All maize growth parameters (average plant height, root and shoot weight) and average cob weight were significantly (*p* < 0.05) higher in the *M. maydis* and *M. phaseolina* double infection compared to the single fungus inoculation. This influence was also reflected in the severity of disease symptoms (Table 6). The double infection treatment reduced disease damage and increased the number of healthy plants from two to three. 

The 2019 repetition supported the 2018 findings of the combined effect of *M. maydis* and *M. phaseolina* on the maize plants’ development and health (Figure 4 and Figure 5, respectively). Similar to the 2018 experiment, in the 2019 experiment, no statistically significant difference between the treatments or the control could be identified during the emergence stage (9 DAS, Figure 4A). At this stage, the sole *M. maydis* inoculation caused higher (statistical difference, *p* < 0.05) values of emergence than the sole *M. phaseolina* inoculation, which resulted in opposite lower values. However, the treatments had similar growth parameters to the control (without any statistical difference) and these differences were later blurred at the sprouting stage (23 DAS, Figure 4B). 

The lack of sufficient irrigation had a minor effect during the early growth period (23 DAS), reflected mainly in the above-ground parts (lower shoot biomass, plant height, and the number of leaves). In the water stress group, the *M. phaseolina* treated plants showed significant elevation in their height and shoot weight, compared to the control. Also, at this watering regime, the double inoculation with *M. maydis* and *M. phaseolina* led to significantly high shoot weight (*p* < 0.05, Figure 4B). Latter, at 58 DAS, the double infection or sole infection of the plants with *M. phaseolina* enhanced their progression from the VT (tasseling) to the R1 (silking) phenological stage (Figure 4C). Drought stress was a crucial factor in this regard, causing all maize plants to be delayed in the VT phase. At the season-ending, 79 DAS, the differences between the treatments were enhanced (Figure 4D). The most prominent positive influence of the double infection (compared to the single *M. maydis* inoculation) was expressed in an increase of 6% in average plant height and 26% in average cob weight. In the 2019 experiment, the benefit of the dual inoculation was most noticeable in the plants’ health evaluation (Figure 5 and Figure 6). Here, the combined infection abolished the severe late wilt symptoms and led to a sharp increase (from 10% to 43%) in the proportion of healthy plants.

### 3.4. The Co-Influence of M. maydis and M. phaseolina on Cotton Plants

In cotton, the molecular tracking of the *M. maydis* and *M. phaseolina* pathogens’ DNA in the plant tissues during the growing season revealed a different picture than in the maize plants at the end of the season. At the sprout stage (37 DAS, Figure 7) and with a balanced water regime, similar to maize, infecting the cotton plants with both pathogens in comparison to infection with each of the pathogens alone led to a 3.9-fold decrease in the levels of *M. maydis* DNA and a 26-fold increase in the levels of *M. phaseolina* DNA in the plant roots. However, at water stress deficiency, the *M. maydis* DNA levels were higher in the double infection treatment than in the single *M. maydis* infection, and the amplification effect of the double infection on *M. phaseolina* DNA levels was less pronounced (Figure 7 upper panels)

At the end of the season (154 DAS), the dual inoculation influence on *M. maydis* DNA was similar to that obtained at 37 DAS under the insufficient water regime, with undetectable DNA levels in the *M. maydis* single inoculation, and low, but still measurable, *M. maydis* DNA levels (5.9*10^-5^) in the double infection treatment (Figure 8). Still, the only observable detection (Mm + Mp) is a result of one repeat out of the 10 repeats (nine repeats resulted in zero detection). However, when the same DNA samples were tested with the *M. phaseolina* primer set (*Mpk*), there were measurable and interesting results that reflect the outcome of the treatments. The highest *M. phaseolina* DNA level was achieved in the single infection treatment, with a 3.7-fold difference from the infection with both pathogens.

Inspecting the treatments’ influence on the cotton plants’ development reveals that the pathogens’ DNA spreading is not always expressed in the plants’ health conditions. At the sprout developmental stage (41 and 57 DAS), only a minor influence of the various infection modes was reflected in changes in growth parameters (Figure 9A–C, Table 7). As expected, the limitation of water supply caused a growth delay.

On the cotton harvest day (154 DAS), the double infection suppressing the influence on *M. phaseolina* DNA spreading (Figure 8) may be the reason for the recovery of plant height and root weight (Table 7). While the maize plants in this two-year study were developing severe late wilt symptoms at an advanced age (70 DAS onwards), the cotton plants showed no evident sign of wilting (Figure 6 and Figure 10), and only a delayed development was noticed in some of the treatments (Table 7). 

The 2019 repetition supported the 2018 findings, with enhancement in all growth parameters and yield productions that were measured (Figure 9D). These results were obtained at 162 DAS under a restricted water regime, while optimal water irrigation led to a lesser difference between the treatments. Supporting the accurate measurements, the 2019 experiment photo (Figure 10) on harvest day (162 DAS) reflects well the recovery in the plants’ development in the double infection treatment in comparison to the *M. phaseolina* single infection treatment.

## 4. Discussion

In Israel, *M. maydis* is considered a major threat to commercial maize fields, while *M. phaseolina* is a primary pathogen causing economic damage in cotton fields. Nevertheless, both pathogens can be found in both crops and may contribute to disease outbreak and spread [10,41]. The current study reveals a complex and competitive interaction between *M. maydis* and *M. phaseolina* species that had been previously unknown. The results of this study strongly suggest an antagonistic co-influence of these two vascular wilt pathogens, which results in restricting *M. maydis* spreading in maize plants and *M. phaseolina* spreading in cotton. This antagonism also has a positive influence on plant growth and yield. 

Antagonistic interactions between phytoparasitic fungi have previously been reported in other host-pathogen species pairs [23], as detailed in the Introduction. In many of those studies, the molecular DNA tracing proved to be an essential research tool [25,27,42]. An example that is very similar to the current research is one of the most documented antagonisms among root pathogens, the suppression of *Cochliobolus sativus* by *Fusarium* (*roseum*) species [42]. In this recent study, a field trial was conducted that included the monitoring of spring wheat plant health and pathogen populations using qPCR. Across field locations, *C. sativus* and *F. pseudograminearum* isolates consistently and significantly reduced one another. However, our data suggest more complex interactions than a simple two-direction antagonism. 

It is interesting to note that infecting maize plants solely with *M. phaseolina* caused a 20% reduction in the number of healthy plants compared to the negative control (from seven plants to five plants, Table 6). Thus, if both pathogens were acting without interfering with each other, their combined effect (50% in the *M. maydis* treatment and 20% in the *M. phaseolina* treatment) should have caused the total wilting of the plants. Instead, the dual inoculation led to some recovery of the plants. Indeed, *M. maydis* reduced *M. phaseolina* DNA in the roots of cotton plants by 4-fold, and *M. phaseolina* reduced *M. maydis* DNA in maize plants stems by 321-fold. Nevertheless, the mixed inoculation also resulted in higher levels of *M. phaseolina* DNA in maize and higher levels of *M. maydis* DNA in cotton. This suggests that the suppression effect of the fungal coexistence is of each pathogen in its primary host, while in its secondary host, the interactions with other phytopathogens can occasionally allow it to flourish. The nature of these interesting relationships and their causes will require further work to resolve. Another question triggered by the results presented here is what is the nature of *M. maydis* interactions with cotton plants? 

It was previously reported that, in cotton, infecting the soil with *M. maydis* caused an increased growth of lateral roots and the appearance of local dark red lesions and shallow cracks on young cotton roots (up to 45 DAS) [12]. Later, as the plants matured, these lesions disappeared, as well as the fungus. Furthermore, *M. maydis* was not recovered from these symptomatic cotton plants [12]. It was recently reported that *M. maydis* infection also affected the root biomass and phenological development of young cotton plants (37 DAS) in a growth chamber [10]. The absence of symptoms in mature cotton plants reported in the above-cited work evokes a new inquisitive question: does this pathogen have a hidden lifestyle as an endophyte in cotton? 

According to the data presented here, this may well be the case. Tracking the effect of *M. maydis* inoculation on cotton plants throughout the whole season in this two-year study supplies a large and consistent dataset. The data presented show that *M. maydis* inoculation at a regular irrigation regime did not affect the cotton growth parameters measured or the yield. On the contrary, in the 2018 experiment, in most measures, this inoculation resulted in higher growth and yield values at the end of the season. In the 2019 experiment, under drought stress, *M. maydis* infection led to a decreased weight and height of the above-ground parts of the cotton plants (without any measurable effect on yield production). This result may suggest that some opportunistic behaviors of this pathogen could exist. This assumption should be tested and established in future studies. 

Formerly, results from random amplified polymorphic DNA (RAPD) analysis showed that *M*. *phaseolina* isolates from maize, cotton, and other crop hosts were differentiated from each other based on the host from which they initially isolated [41,43]. Moreover, isolates within each host that have different sensitivities to chlorate (a powerful oxidizer) also distinctly separated [41,43]. The researchers’ studies of a field with a long history of constant crop rotation led them to suggest that host specialization developed in this fungus. However, host specialization in *M*. *phaseolina* appears to occur with maize, but not with sorghum, cotton, or soybean, as indicated by the results of cross-inoculation experiments [41]. The colonization of maize roots was significantly higher with maize isolates than with isolates from other hosts. Since *M. maydis* distribution is limited to about only eight countries and is considered to be an exotic pathogen in most of the world, the above-mentioned knowledge about its host specialization and chlorate sensitivity is absent. It was only recently reported that the *M. maydis* host range is wider than we had thought [10]. 

It would be very interesting and economically significant to study the differentiation between *M. maydis* populations originating from different host species. Such studies, which were already carried out in maize in Egypt, had shown that isolates of *M. maydis* differ in pathogenicity, morphology, and route of infection [44]. Most importantly, it was reported in Egypt and Spain that this pathogen could undergo pathogenic changes that can result in highly virulent strains [34,45]. The Egyptian isolates of *M. maydis* were classified into four clonal lineages, which revealed not only diversity in virulence and colonization ability on maize but also fascinating competitiveness relationships [34]. The most virulent lineage, when tested alone, was the least competitive on susceptible maize accessions when incorporated into the soil as a component of the mixed inocula of all four lineages. In contrast, one of the less virulent lineages dominated (70% of infections) and appeared to be the most competitive. 

Thus, it was suggested [34] that the most virulent strain may have an advantage in this interspecies competition when the field is seeded with resistance maize cultivar. However, in fields in which susceptible maize hybrids are cultivated, the other less virulent *M. maydis* strains may become more prevalent, and the most virulent lineage may be relatively rare. It should be taken into consideration that environmental aspects (biological and physical) are involved. *M. maydis* strains may differentiate due to their ability to grow either in the soil or in the host plant and due to their sensitivity to other soil microbes or their metabolic products.

The reciprocal suppression that was documented between *M. maydis* and *M. phaseolina* species in this study raises the importance of inspecting the effectiveness of preventing treatments while considering the pathogen specialization and the circumstances under which several pathogens are involved. To emphasize this, in spring 2018, a commercial field (Neot Mordechai field, Hula Valley, Upper Galilee, northern Israel) was planted with the sensitive maize cultivar Prelude cv. This field had a long history of late wilt incidences [5] and was therefore protected by a recently developed and highly effective chemical method [6]. Despite the use of the potent commercial mixture of Azoxystrobin and Difenoconazole (Syngenta, Basel, Switzerland), at the end of the season, only meager yields were achieved. The yield reduction was most probably the consequence of another fungal disease caused by *F. verticillioides* and *F. oxysporum*, which led to dehydration and yield loss (Ofir Degani et al., unpublished results). These Fusarium species are indeed universal and important vascular wilt disease-causing fungal pathogens [46]. To support this conclusion, in other plants in that field that were not affected by the *Fusarium* spp. disease, the chemical treatment almost wholly abolished any sign of late wilt disease.

It has now become evident that optimal water supply can reduce late wilt disease in maize. Indeed, our past experience [6,30,33] and a review of the literature support the conclusion that high water potential is one of the most important agrotechnical aims for restricting late wilt disease progression [17,47,48,49]. The results presented in the current study are in line with this conclusion, with nearly 10 times higher *M. maydis* DNA in drought-stressed maize sprouts (37 DAS, Figure 2). Interestingly, drought pressure also caused about three times higher *M*. *phaseolina* DNA in young maize plants, but both pathogens did not flourish under a restricted water regime in cotton (Figure 6) compared to regular irrigation conditions. A possible explanation is that the applied restricted water regime did not produce drought pressure as expected. Indeed the cotton plants in the decrease water irrigation pots were healthy and had an apparent healthy development. 

## Figures and Tables

**Figure 1 microorganisms-08-00249-f001:**
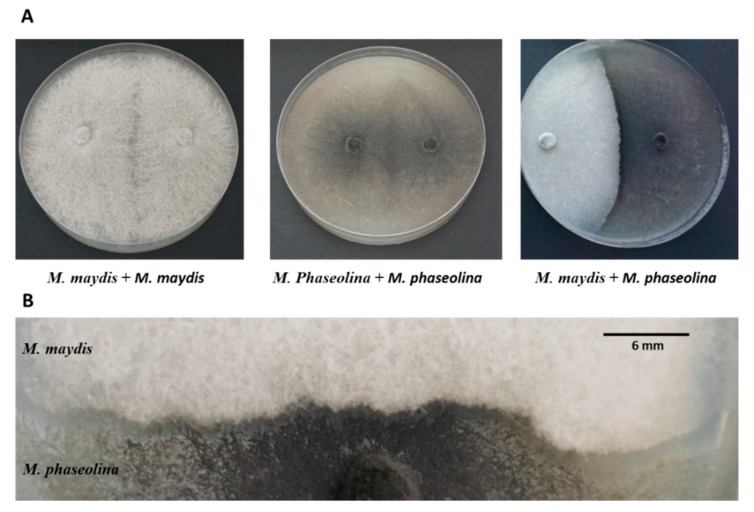
Plate confrontation assays. The plate assay to identify interactions between *Magnaporthiopsis maydis* and *Macrophomina phaseolina* in a rich potato dextrose agar (PDA) culture media. (**A**) Left panel—*M. maydis* was seeded on both sides of the Petri dish. Center panel—*M. Phaseolina* was seeded at both poles of the dish. Right panel—the two fungi were seeded opposite each other, *M. maydis* on the left, and *M. phaseolina* on the right. (**B**) Magnification of the confrontation line between *maydis* (upper white hyphae) and *M. phaseolina* (lower dark hyphae). The two colonies formed a clear line between them. Images were taken after six days of growth at 28 °C ± 1 in the dark. Scale bar represents 6 mm.

**Figure 2 microorganisms-08-00249-f002:**
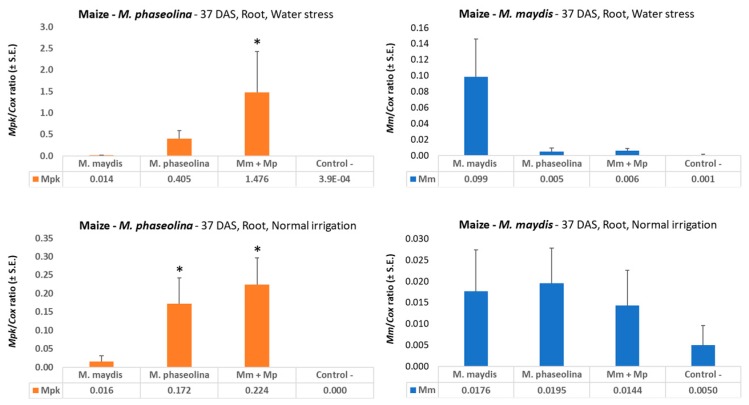
qPCR diagnosis of *M. maydis* and *M. phaseolina* in maize sprouts. A full-growth season pot experiment under field conditions was conducted in the spring and summer of 2019 with a late-wilt-susceptible maize genotype prelude. The quantitative real-time polymerase chain reaction (qPCR) was used to identify the ability of the pathogens, alone or in mixed inoculation, to infect and colonize maize root tissues 37 days after sowing (DAS). Half of the treatments received reduced irrigation (water stress, upper panel, see Table 1). The control is soil taken from a nearby field with minor levels of *M. maydis* or *M. phaseolina* infestation. The y-axis values are *M. maydis* relative DNA (*Mm*) or *M. phaseolina* relative DNA (*Mpk*) abundance normalized to the cytochrome c oxidase (*Cox*) DNA. Vertical upper bars represent the standard error of the mean of 10 replications (pots, each containing one plant). When existing, significance from the control is indicated as * = *p* < 0.05.

**Figure 3 microorganisms-08-00249-f003:**
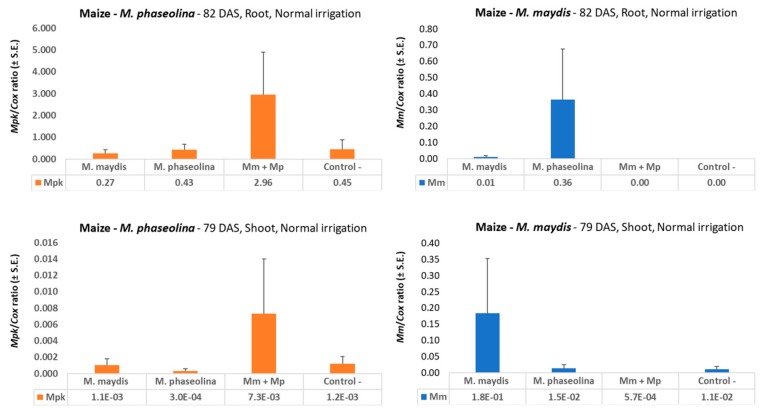
qPCR diagnosis of *M. maydis* and *M. phaseolina* in maize at the end of the season. The experiment is described in Figure 2. *M. maydis* relative DNA (*Mm*) or *M. phaseolina* relative DNA (*Mpk*) abundance normalized to the cytochrome c oxidase (Cox) DNA was evaluated at 82 DAS (25 days after fertilization, DAF) in roots in the 2018 experiment (upper panels) and 79 DAS (25 DAF) in shoots in the 2019 experiment (lower panels). Vertical upper bars represent the standard error of the mean of 10 replications (pots, each containing one plant).

**Figure 4 microorganisms-08-00249-f004:**
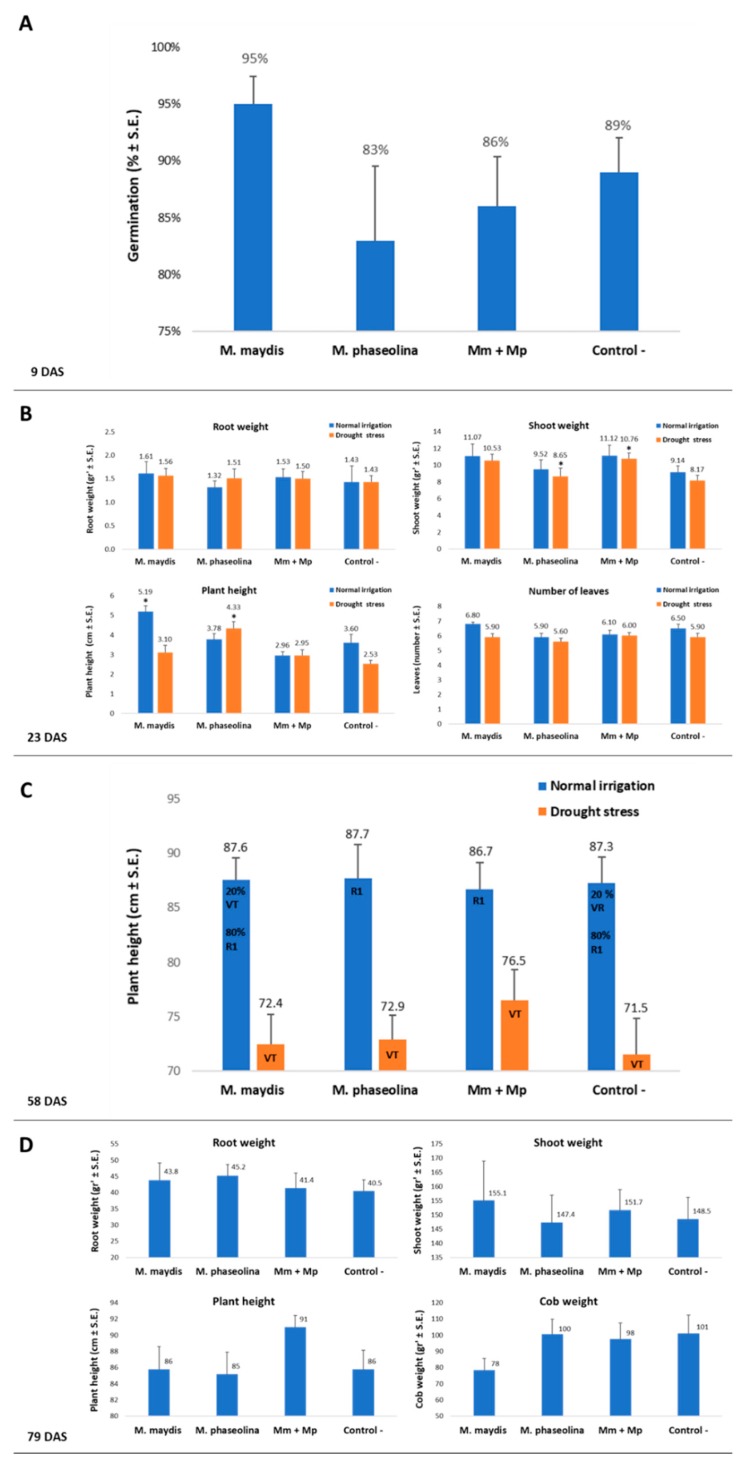
Symptoms evaluation and yield assessment in maize. Plant developmental values and yield assessment was made for the plants in the 2019 experiment described in Figure 2. Sprout emergence, 9 DAS (**A**), phenological stage, 23 DAS (**B**), plant height, 58 DAS (**C**), and growth and yield assessment 79 DAS and 22 DAF (**D**) were determined after the inoculation of the plants with *M. maydis* and *M. phaseolina*, alone or in mixed inoculation. Vertical upper bars represent the standard error of the mean of 10 replications (pots, each containing one plant from 23 DAS onwards). When existing, significance from the control (untreated) is indicated as * = *p* < 0.05.

**Figure 5 microorganisms-08-00249-f005:**
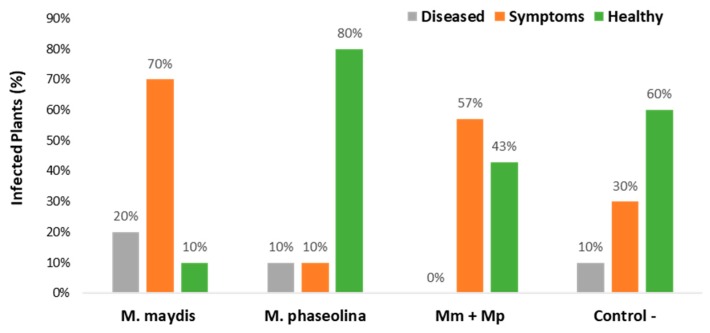
Maize wilt assessment carried out 79 days after sowing. Wilting percentages of the 2019 experiment described in Figure 2 were determined 21 DAF. Plants were classified as “healthy” when no apparent signs of dehydration could be identified and as “symptoms” when the upper leaves’ color started to alter to light-silver and then to light-brown, rolling inward from the edges of the leaf. When the whole plant was presenting severe dehydration with light-brown color spreading to all of its parts and its cobs tilted down, it was classified as “diseased.” Percentages are the mean of 10 replications.

**Figure 6 microorganisms-08-00249-f006:**
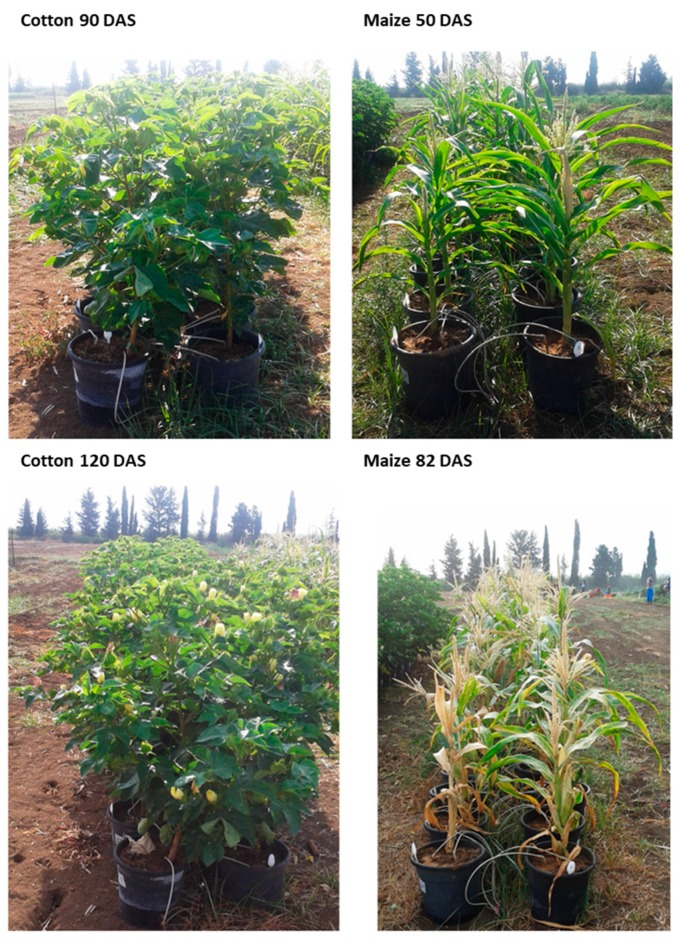
Photograph of the maize and cotton plants in the 2018 experiment. The experiment at the vegetative growth stage, 50 and 90 DAS in maize and cotton, respectively (upper panels), and at the end of the season, 82 and 120 DAF in maize and cotton, respectively (lower panels).

**Figure 7 microorganisms-08-00249-f007:**
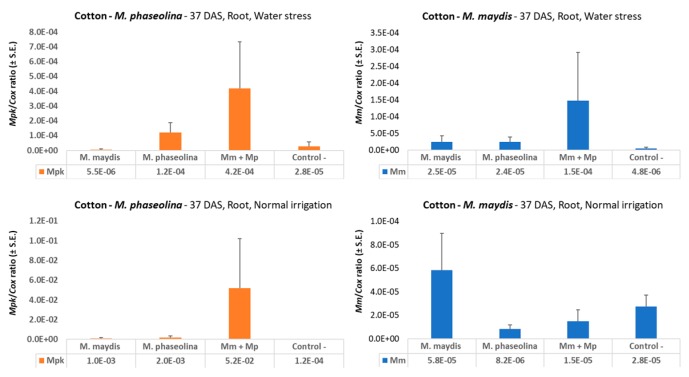
qPCR diagnosis of *M. maydis* and *M. phaseolina* in cotton sprouts. The full-growth season pot experiment under field conditions was conducted in the spring and summer of 2019. The charcoal-rot-susceptible cotton genotype Pima cotton, Goliath cv. was chosen for this experiment. The qPCR was used to identify the ability of the pathogens, alone or in mixed inoculation, to infect and colonize the cotton root tissues 37 days after sowing (DAS). The control is a soil taken from a nearby field with minor levels of *M. maydis* or *M. phaseolina* infestation. The y-axis values are *M. maydis* relative DNA (*Mm*) or *M. phaseolina* relative DNA (*Mpk*) abundance, normalized to the cytochrome c oxidase (*Cox*) DNA. Vertical upper bars represent the standard error of the mean of 10 replications (pots, each containing one plant).

**Figure 8 microorganisms-08-00249-f008:**
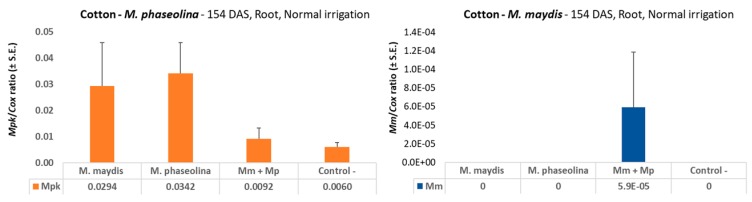
qPCR diagnosis of *M. maydis* and *M. phaseolina* in cotton at the end of the season (154 DAS). The experiment was conducted in 2018. The control is a soil taken from a nearby field with minor levels of *M. maydis* or *M. phaseolina* infestation. *M. maydis* relative DNA (*Mm*) or *M. phaseolina* relative DNA (*Mpk*) abundance was normalized to the cytochrome c oxidase (Cox) DNA. Vertical upper bars represent the standard error of the mean of 10 replications (pots, each containing one plant).

**Figure 9 microorganisms-08-00249-f009:**
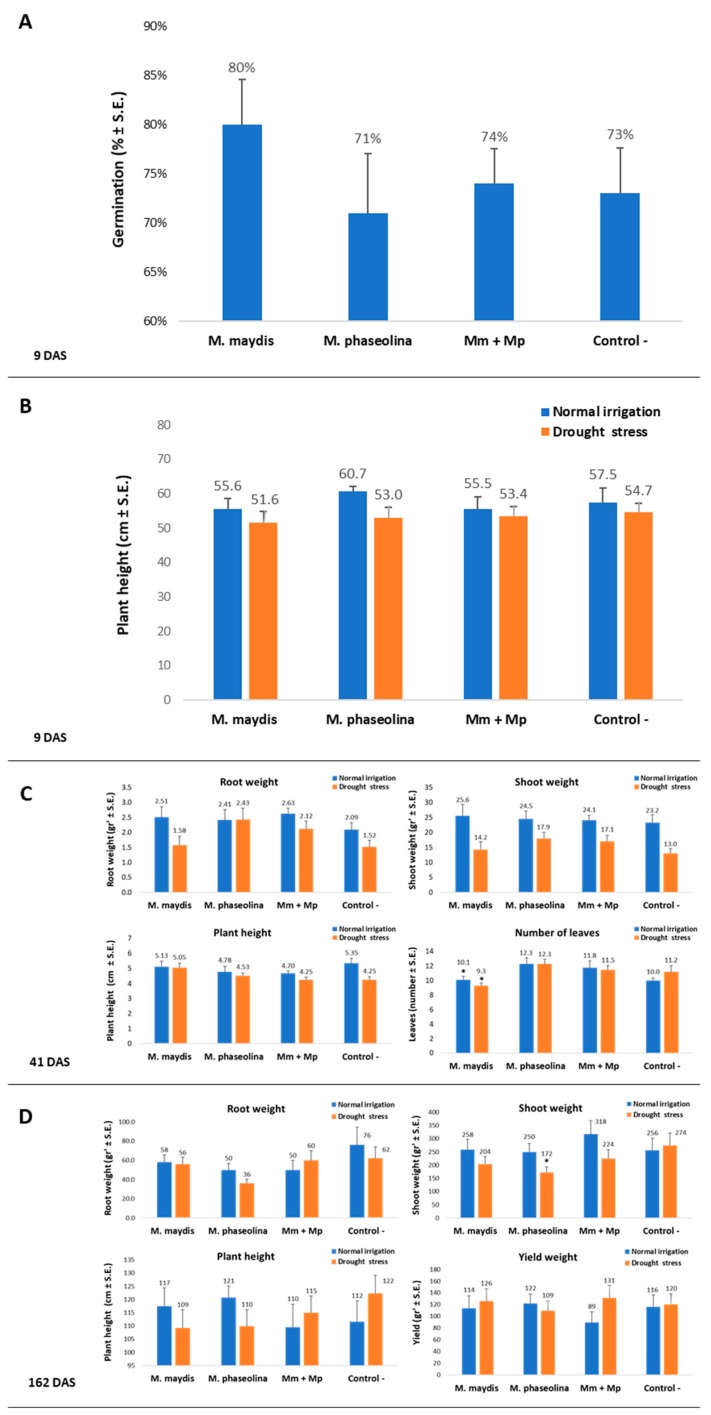
Symptoms evaluation and yield assessment in cotton. Plant developmental values and yield assessment was made for plants in the 2019 experiment described in Figure 7. Sprout emergence (**A**) and height, (**B**) 9 DAS, phenological development, 41 DAS (**C**), and growth and yield assessment, 162 DAS (**D**), were determined after the inoculation of the plants with *M. maydis* and *M. phaseolina*, alone or in mixed inoculation. Vertical upper bars represent the standard error of the mean of 10 replications (pots, each containing one plant 41 DAS onwards). When existing, significance from the control (untreated) is indicated as * = *p* < 0.05.

**Figure 10 microorganisms-08-00249-f010:**
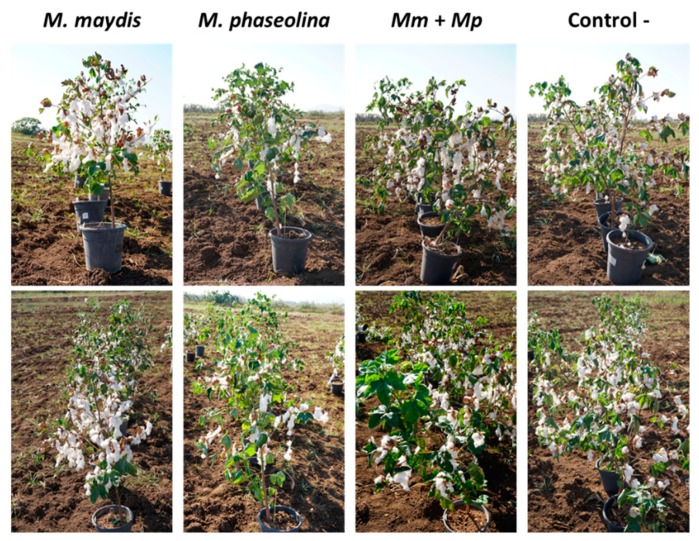
Photograph of the cotton plants under a restricted water regime, in the 2019 experiment (described in Figure 7). The field was photographed 162 DAS. A decreased development of the plants in the *M. phaseolina* inoculation pots can be seen.

**Table 1 microorganisms-08-00249-t001:** Water irrigation regime during the full-growth season pot experiments (2018 and 2019).

Year	Irrigation Regime	Maize		Cotton	
		Age (DAS) ^2^	Irrigation (L per pot/2 day)	Age (DAS)	Irrigation (L per pot/2 day)
2018	Normal ^1^	0–6 ^3^	5.4	0–28 ^3^	5
		7–82	2.7	29–154	2.5
2019	Normal ^1^	0–48	4	0–162	4.8
		49–79	6		
	Drought pressure	0–48	2	0–162	4
		49–79	4		

^1^ A healthy irrigation regime was conducted according to the Israel Ministry of Agriculture Consultation Service (SAHAM) growth protocol. The total amount of water for each season was estimated to be 600 mm using the Penman method. ^2^ DAS—days after sowing. ^3^ Until the plants’ first emergence above the ground surface, the pots were irrigated to a near saturation capacity in order to allow for proper germination and initial development.

**Table 2 microorganisms-08-00249-t002:** Meteorological data for the 2018 and 2019 experiments.^1.^

	2018		2019	
Environmental Condition	Maize 24/05-14/08	Cotton 02/05-03/10	Maize 21/05-08/08	Cotton 21/05-30/10
Temperature (°C)	27.1 ± 5.2	26.8 ± 5.4	27.3 ± 5.7	26.5 ± 5.7
Humidity (%)	60.4 ± 18.0	59.9 ± 19.1	57.2 ± 19.6	60.5 ± 19.7
Soil temp. top 5 cm (°C)	28.9 ± 7.4	31.7 ± 8.9	33.8 ± 12.1	32.1 ± 19.7
Radiation (*W*/*m*^2^)	328.9	302.4	355.6	289.5
Precipitation (mm)	9.9	31.4	0.5	53.8
Evaporation (mm)	724.2	1257.04	669.6	1201.6

^1^ Data (average ± standard deviation) according to the Israel Northern Research and Development meteorological station data.

**Table 3 microorganisms-08-00249-t003:** Primers for Magnaporthiopsis maydis and Macrophomina phaseolina detection.

Pairs	Primer	Sequence	Uses	Amplification	References
Pair 1	A200a-for A200a-rev	5′-CCGACGCCTAAAATACAGGA-3′ 5′-GGGCTTTTTAGGGCCTTTTT-3′	Target gene	200 bp *M. maydis* species-specific fragment	[5]
Pair 2	MpKFI MpKRI	5′-CCGCCAGAGGACTATCAAAC-3′ 5′- CGTCCGAAGCGAGGTGTATT-3′	Target gene	300–400 bp *M. phaseolina* species-specific fragment	[31]
Pair 3	ITS1 ITS4	5′- TCCGTAGGTGAACCTGCGG -3′ 5′- TCCTCCGCTTATTGATATGC -3′	Target gene	Target region for the identification of fungi species	[39]
Pair 4	COX-F COX-R	5′-GTATGCCACGTCGCATTCCAGA-3′ 5′-CAACTACGGATATATAAGRRCCRR AACTG-3′	Control	Cytochrome c oxidase (*COX*) gene product	[37,40]

**Table 4 microorganisms-08-00249-t004:** Maize average growth indices at 34 DAS ^1^ as a result of infection with *M. maydis* and *M. phaseolina* (the 2018 experiment).

Growth Parameter	*M. maydis*	*M. phaseolina*	*M. maydis +* *M. phaseolina*	Control ^2^
Emergence (%)	64 ± 9.3	72 ± 5.3	70 ± 6.8	76 ± 5.8
Height (cm)	22.0 ± 1.65	21.4 ± 1.34	21.5 ± 1.41	19.9 ± 1.20
Root weight (g)	3.7 ± 0.80	3.5 ± 0.53	4.0 ± 0.36	3.0 ± 0.52
Shoot weight (g)	26.4 ± 4.6	24.3 ± 4.2	24.5 ± 3.7	18.4 ± 2.6

^1^ DAS—days after sowing. Values represent an average of 10 replications ± Standard error. No statistically significant (*p* < 0.05) difference was identified between the treatments and the control. ^2^ Control—soil from a nearby field with minor levels of *M. maydis* or *M. phaseolina* infestation.

**Table 5 microorganisms-08-00249-t005:** Maize average growth indices at 82 DAS (25 DAF ^1^) as a result of infection with *M. maydis* and *M. phaseolina* (the 2018 experiment).

Growth Parameter	*M. maydis*	*M. phaseolina*	*M. maydis +* *M. phaseolina*	Control ^3^
Height (cm)	85.4 ± 4.9	94.9 ± 5.0	111.7 ± 5.4 ^2^	109.2 ± 3.8
Root weight (g)	40.5 ± 8.3	82.9 ± 32.1	59.0 ± 9.1 ^2^	54.9 ± 10.6
Shoot weight (g)	174.5 ± 22.6	214.2 ± 34.9	245.1 ± 13.0 ^2^	201.5 ± 24.6
Cob weight (g)	91.4 ± 16.0	112.7 ± 16.0	170.7 ± 22.9 ^2^	125.1 ± 16.6

^1^ DAF—days after fertilization. ^2^ Significant difference (*P* < 0.05) from the sole *M. maydis* infection treatment and the control. ^3^ Control—soil from a nearby field with minor levels of *M. maydis* or *M. phaseolina* infestation.

**Table 6 microorganisms-08-00249-t006:** Maize dehydration assessment at 82 DAS following infection with *M. maydis* and *M. phaseolina* (the 2018 experiment).

Treatment	Healthy	Mild Symptoms	Symptoms	Wilted
*M. maydis*	20%	30%	10%	40%
*M. phaseolina*	50%	30%	10%	10%
*M. maydis* + *M. phaseolina*	30%	10%	20%	40%
Control ^1^	70%	30%	0%	0%

^1^ Control—soil from a nearby field with minor levels of *M. maydis* or *M. phaseolina* infestation.

**Table 7 microorganisms-08-00249-t007:** Cotton average growth indices at 57 DAS (at flowering) and at 154 DAS (at harvest) as a result of infection with *M. maydis* and *M. phaseolina* (the 2018 experiment).

Days after Sowing	Growth Parameter	*M. maydis*	*M. phaseolina*	*M. maydis + M. phaseolina*	Control
57	Height (cm)	46.7 ± 3.0	53.6 ± 1.2	48.3 ± 2.1	50.0 ± 1.6
	Root weight (kg)	12.0 ± 1.3	11.5 ± 1.2	8.5 ± 0.9	10.8 ± 1.1
	Shoot weight (kg)	74.1 ± 11.2	72.1 ± 7.5	54.0 ± 7.4	62.8 ± 4.6
	Number of leaves	20.6 ± 2.7	21.6 ± 1.3	17.6 ± 1.1	18.7 ± 1.1
154	Height (cm)	116.4 ± 3.7	105.3 ± 4.1	117.2 ± 4.6^1^	108.9 ± 6.8
	Root weight (kg)	0.14 ± 0.02	0.10 ± 0.01	0.15 ± 0.02^1^	0.14 ± 0.02
	Shoot weight (kg)	0.50 ± 0.06	0.39 ± 0.07	0.62 ± 0.12	0.39 ± 0.09
	Average crop weight (kg)	0.25 ± 0.03	0.19 ± 0.03	0.23 ± 0.04	0.19 ± 0.03
	Number of seeds	15.1 ± 0.59	15.4 ± 0.42	14.7 ± 0.99	13.7 ± 0.53

^1^ Significant difference (*p* < 0.05) from the sole *M. phaseolina* infection treatment.

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
