# Peer review of "Interactions between Magnaporthiopsis maydis and Macrophomina phaseolina, the Causes of Wilt Diseases in Maize and Cotton"

_microorganisms, 2020, doi:10.3390/microorganisms8020249_

Round 1
Reviewer 1 Report
Major comments:
Regarding infested field soil-based pathogenicity test, how did authors ensure that each pot received equal amount of inoculum for both the fungi? Did they check whether that soil contained any other pathogenic fungi or microorganisms? Did they quantify the presence of both the fungi (by total DNA isolation from soil and subjecting it for qPCR with M. maydis- and M. phaseolina-specific primers) in soil before using it for path trials? To examine whether the control soil was contaminated by M. maydis or M. phaseolina, authors should have carried out a similar qPCR-based diagnosis. Why they did not used simply autoclaved soil for path trials?
Regarding qPCR assay, did authors conduct any inhibitor identification assay to check the presence of PCR inhibitors in the DNA samples (being used for qPCR)? This can be done by designing an artificial template and its corresponding primers. This artificial template is needed to be spiked into each sample and amplified using specific primers. In case of absence of qPCR inhibitors, Ct values will be similar for all the tested samples and positive control. This is known as SPUD assay.
I am also not satisfied with the way of data representation for qPCR results. I think it would be better to mention the fungal burden/DNA abundance for each fungal combination on Y-axis (amount of normalized fungal DNA). There should also be a standard curve, based on which fungal burden for each fungus can be calculated.
Minor comments:
Line 260: latter or former?
Line 267: Why calling this anastomosis between these two fungi "antagonism plate assay" especially when there are no perceptible antagonistic interactions can be seen here. I think describing it as plate confrontation assay is sufficient.
Table 3. where are the brackets?
Suggestion:
Fungal viruses are omnipresent in all the major groups of fungi. In several instances, their infections were found to be associated with different types of phenological changes in their hosts including growth, sporulation, host fitness and virulence. It would be interesting to investigate roles of such mycoviruses in these two fungi and their role in host pathogenicity.
Author Response
Responses to Reviewer 1’s comments
We would like to express our appreciation to the reviewer for the important and helpful corrections, suggestions and advice. This contribution significantly improved the manuscript. Thank you.
Major comments:
Regarding the infested field soil-based pathogenicity test, how did authors ensure that each pot received an equal amount of inoculum for both the fungi?
The reviewer is correct; this is indeed an important issue that should be clarified. Since we aim our experiments to simulate field conditions, they were performed on naturally infested field soil. The field soil taken for the experiment proved to be infested with both M. maydis and M. phaseolina pathogens (Degani et al., PloS One 2018, 13, e0208353, and Cohen et al., 5th Conference of the Israel Society of Crop and Vegetable Sciences, 2018). This information was already stated in the text (lines 175-178).
The reason for using pots with naturally infested soil in the field open air instead of sowing the plants directly in the field soil was to allow enhancing the soil inocula to achieve high and equable infection as much as possible, and for better control of the water regime.
To elaborate on this, the pathogenicity trials cannot rely on natural soil infestation alone, which can lead to highly variable results. Even in heavily infested fields, the spreading of the pathogen is not uniform. The pathogen is scattered in small quantities in the soil, and the disease spreading is not uniform in the field (see, for example, Degani et al., Methods for studying Magnaporthiopsis maydis, the maize late wilt causal agent. Agronomy 2019, 9, 181, Figure 1).
Thus, we used a complementary inoculum to ensure that the inoculation of the plants would be as high and uniform as possible. Together with this, knowing that some variations in the amount of inoculum is predicted, we repeated the whole experiment twice (in two consecutive years) and conducted each treatment in 10 repeats. This experimental design aimed at minimizing those differences. Overall, we achieved similar results in the two experiment repetitions and within each treatment results.
The following explanation was added to the Materials and Methods text (lines 140-147): “The experiments’ aim was to simulate field conditions, and the reason tofor useing pots (positioned in the open-air field open-air) with naturally infested soil, instead of sowing the plants directly to the field soil was to allow enhancing the soil inocula in order to achieve as much as possible high and equable infection as much as possible, and for better controlling of the water regime. To elaborate on this, pathogenicity trials cannot rely on natural soil infestation alone, which can lead to highly variable results. Even in heavily infested fields, the spreading of the pathogen is not uniform. The pathogen is scattered in small quantities in the soil, and the disease spreading is not uniform in the field.”
Did they check whether that soil contained any other pathogenic fungi or microorganisms?
Indeed, the soil most certainly contains other microorganisms, some of which may be pathogenic fungi. However, the non treated negative control group (soil taken from a nearby field that had no or very low M. maydis or M. phaseolina infestation) and the experimental groups are proving that the unique disease symptoms outcome and the molecular targeting of both pathogens’ DNA spreading inside the host tissues are the consequences of the treatments. The possible influence of other microorganisms in the soil in this regard is very interesting and should be the focus of a follow-up work that will examine this question thoroughly.
Did they quantify the presence of both the fungi (by total DNA isolation from the soil and subjecting it for qPCR with M. maydis - and M. phaseolina-specific primers) in the soil before using it for path trials?
There is no molecular soil assay currently available for M. maydis detection (DNA isolation and identification PCR or qPCR). In fact, we are trying to develop such an assay (with PCR and a commercial soil kit assay), but we have learned that this is a complicated task. One of the reasons is that, as mentioned above, the pathogen is scattered in small quantities in the soil, and the disease spreading is not uniform in the field. The use of susceptible maize cultivar planted in infected soil, as demonstrated in the current work, may be considered as a bioassay in which the plant is used to isolate and enrich the pathogen from the soil, and the qPCR diagnostic is used as a sensitive method to measure its prevalence. We recently completed work on such a bioassay and are preparing it for publication.
To examine whether the control soil was contaminated by M. maydis or M. phaseolina, authors should have carried out a similar qPCR-based diagnosis. Why they did not use simply autoclaved soil for path trials?
The difference between the method presented in this work and the use of artificial autoclaved soil inoculation (adding inoculum to non-contaminated soil) was discussed in our previous work (see Degani et al., Evaluating Azoxystrobin seed coating against maize late wilt disease using a sensitive qPCR-based method. Plant Disease [2019], 103 [2] 238-248). Deliberately infecting healthy soil with M. maydis was incapable of causing significant symptoms in mature plants (aged 70 days or more).
It should also be taken into consideration that autoclaving and sterilizing the soil may alter its properties, and that such soil is most likely to be different from natural field soil. We carried out work of this type with maize and cotton plants (up to the age of 40 days) inoculated with M. maydis or M. phaseolina and planted in non-contaminated soil under controlled conditions. However, we decided not to include it in the current report since the results, although interesting and valuable, were very different from the results we had obtained using naturally infested soil in open-air, which best matches the situation in commercial fields.
The control group used in both years’ experiments was designed to be as similar as possible to the inoculation treatments in order to provide a reference point. As detailed in the text (lines 152-154), the negative control in the experiments was soil taken from a nearby field that had no history of M. maydis or M. phaseolina infestation, and if such an infestation did exist, it was assumed to be very low.
Regarding qPCR assay, did authors conduct any inhibitor identification assay to check the presence of PCR inhibitors in the DNA samples (being used for qPCR)? This can be done by designing an artificial template and its corresponding primers. This artificial template is needed to be spiked into each sample and amplified using specific primers. In the case of the absence of qPCR inhibitors, Ct values will be similar for all the tested samples and positive control. This is known as the SPUD assay.
This is good advice. Indeed, a similar efficiency was assumed since we used the same protocol for all samples examined in this work and paid full attention to all of the details (the sample DNA extraction and purification, the mixture ingredient concentrations and amounts, the qPCR conditions, etc.). The SPUD assay is indeed one option for the identification of inhibitors that may be present in our DNA samples. The assay is particularly useful when targets are present at low copy numbers, making dilution of the sample impractical, or to avoid false negatives and lend greater confidence to reporting on data based on a negative result. However, this may not the case here. The qPCR method we used is very sensitive and capable of detecting variations in the amount of the pathogens’ DNA inside the host plant tissues, with a million-fold difference (see, for example, Degani et al., PloS One 2018, 13, e0208353). When the tissue was infected, the qPCR was successful in identifying and quantifying the pathogens’ DNA. In some of the plants, the pathogen did not succeed in causing infection, which was probably the reason for the negative result.
The use of a SPUD assay may indeed improve our ability to determine this with greater confidence. Unfortunately, we did not perform an inhibitor identification assay, and we cannot repeat the qPCR analysis of the samples while adding the assay at this stage. However, this is not an obligatory procedure. The specific M. maydis qPCR detection was just recently validated, approved and published (Degani et al., Plant Disease, 2019, 103, 238-248). We used this qPCR method repeatedly, without the SPUD assay, in several additional works (see Degani et al., O. Methods for studying Magnaporthiopsis maydis, the maize late wilt causal agent. Agronomy 2019, 9, 181; Dor, S. and Degani, O., Uncovering the host range for maize pathogen Magnaporthiopsis maydis. Plants 2019, 8). The same protocol (without the SPUD assay) with some adjustments is used worldwide in the scientific community for a similar purpose (identifying pathogens DNA inside host tissues).
I am also not satisfied with the way of data representation for qPCR results. I think it would be better to mention the fungal burden/DNA abundance for each fungal combination on the Y-axis (amount of normalized fungal DNA). There should also be a standard curve, based on which fungal burden for each fungus can be calculated.
Indeed, fungal DNA abundance for each fungal combination can be presented on the Y-axis, but the pathogens’ relative DNA abundance normalized to the cytochrome c oxidase (Cox) DNA (or to other “housekeeping” reference genes) is also common and widely used in the scientific community. As mentioned above, the specific qPCR detection was just recently validated, approved and published in several works. The same methodology for presenting the qPCR results has already been approved by leading scientific journals that had published our recent works (Plant Disease, Plos One, Agronomy, Plants). The relative quantification of the target M. maydis or M. phaseolina fungal DNA was calculated according to the ΔΔCt model, as detailed in the text (lines 253-255 and the relevant figure legends).
Minor comments:
Line 260: latter or former?
Right, corrected to “former.”
Line 267: Why calling this anastomosis between these two fungi “antagonism plate assay” especially when there are no perceptible antagonistic interactions can be seen here. I think describing it as a plate confrontation assay is sufficient.
Corrected as per the reviewer’s advice. The sentence is now written: “The plate assay to identify interactions between Magnaporthiopsis maydis and Macrophomina phaseolina in rich potato dextrose agar (PDA) culture media.” (lines 280-282).
Table 4. where are the brackets?
Corrected to: Values represent an average of 10 replications ± Standard error (line 328).
Suggestion:
Fungal viruses are omnipresent in all the major groups of fungi. In several instances, their infections were found to be associated with different types of phenological changes in their hosts including growth, sporulation, host fitness and virulence. It would be interesting to investigate the roles of such mycoviruses in these two fungi and their role in host pathogenicity.
Thank you for this intriguing idea! This will be an excellent and challenging topic for a continuation work.
Reviewer 2 Report
This study presents intriguing evidence for an interaction between Macrophomina phaseolina and Magnaporthiopsis maydis in maize and cotton, which could be of practical significance for disease management. The overall approach involves a well thought out combination of molecular detection of the pathogens and measurements of plant characteristics. The paper is clearly written in good English, but some word choices could be improved (notably “outburst” = outbreak or development; “session” = season; “strict water” = restricted water; “seeding” = sowing) and a number of typographical errors should be corrected. Unfortunately, there are substantial problems with the analysis and presentation of the results.
qPCR
Line 242 states that each sample was tested four times, and line 248 states that amplifications were performed in triplicate. Does this mean there were twelve amplifications for each sample? Line 248 states that the ΔΔCt method was used, but in the figures the data are all presented as ratios of test gene/control gene, which would be calculated from ΔCt. ΔΔCt would give fold differences. Which is correct? Labelling of the Y axes in the figures is incorrect. The Y axis labels show control/test ratios, but these should be test/control ratios. Line 284 states that there were no statistically significant differences but the results are written as though the differences were real. They do look as though some of them might be real. What statistical test was used? In Figure 3, with normal irrigation why is there maydis DNA in plants infected with M. phaseolina but none in plants inoculated with M. maydis? Similarly, in Figure 7 why is there apparently much more M. maydis DNA in plants inoculated with M. phaseolina than in plants inoculated with M. maydis? Line 393 states that under water stress maydis DNA levels were lower in the double infection treatment than in the single M. maydis infection, but according to Figure 6 they were higher.
Statistical analysis
Lines 146 and 253 state that the experiments were performed in a completely randomised design but Figures 9 and 10 show the plants in lines for each treatment. Were they moved into lines for the photographs? Why were t tests used for comparisons? ANOVA (one-way or two-way, depending on the experiment) should be more informative and reliable. Unless there is a good reason to use t tests, the data should be reanalysed. Differences among proportions of plants showing different severity levels should be tested statistically. In Fig. 5, why are the proportions for Mm+Mp 57% and 43% if there were 10 plants? The figure legend says “Percentages are the mean of 10 replications”. Presumably this means proportions of the 10 replicate plants. Line 334 and following states that the double infection increased the number of healthy plants from 20% to 30% and infecting plants solely with phaseolina reduced numbers of healthy plants from 70% to 50%. These are differences of one and two plants, respectively, so probably not significant. Similarly, line 368 states that combined infection abolished severe symptoms, yet this is based on a reduction from two plants with severe symptoms to none. Line 357 states that sole maydis inoculation caused higher values of emergence. Was this tested statistically? In panel A of Figures 4 and 8, are the means and standard errors for the five seeds per pot? The figure legend doesn’t say.
Other points
Line Comment
197 What part of the stem was used (e.g. slice, outside, inside) and how much? Were the fungi visibly present? This description should be moved to section 2.5 to go with the description of DNA extraction from roots.
228 Cross-section from every root of the plant? What part of the root? Were the fungi visibly present?
244 The standard should be identified as the COXI gene coding for subunit 1 of cytochrome c oxidase. For clarity, it should be stated that the amplified gene is from the host plant.
313 In Table 3, are the errors standard errors?
329 “Profound” seems to overstate the effects.
347 Why are there two asterisks with the values in the dual inoculation column? Does one refer to each comparison? Differences in height and root weight between dual infection and control do not seem to be significant.
442 What treatment or treatments does Figure 9 show?
447 What is the difference between the top and bottom rows of pictures in Figure 10?
473 How are these percentage reductions calculated? It is not possible to have a reduction of more than 100%.
501 Chlorate cannot be a nitrogen source as it contains no nitrogen!
Reviewer 3 Report
Manuscript prepared correctly, research methodology correct and very detailed. It requires minor corrections.
Line 39 correct “late wilt,” to “late wilt"
Line 59 "lupini" should be in italics
Line 117 there is a reference to a table that is not in the manuscript at all
Paragraph 130-140 This part contains too much results. Descriptions of this type should be included in the "Results" section. Here I propose to include the methodology.
Table 1 "6-82" should be "7-82"?
Lines 252-255 Please specify which statistical software was used
Lines 301-312 there is no need to repeat information from the methodology
Lines 334-339 such comments are characteristic of the "Discussion" section, they should not be included in the "Results" section
The manuscript contains a lot of figures. Some can be removed, e.g. figs 9, 10
Please try to combine tables 6 and 7
Author Response
Responses to Reviewer 2’s comments
We thank the reviewer for investing time and effort, which contributed to this manuscript. The helpful and important remarks and suggestions improved this scientific paper and made it more accurate, clear and focused.
Line 39 correct “late wilt,” to “late wilt”
Corrected as advised.
Line 59 “lupini” should be in italics
Corrected as advised.
Line 117 there is a reference to a table that is not in the manuscript at all
The reference to the table was removed and replaced by the phrase “as we will elaborate below.”
Paragraph 130-140 This part contains too many results. Descriptions of this type should be included in the “Results” section. Here I propose to include the methodology.
The reviewer is correct. The following section was transferred to the Results section (Lines 268-271): “In this assay, a dominant fungus will usually continue to grow behind the confronted line while covering the other fungi. The results of this assay were compared to the growth rate of each fungus alone under the same conditions in order to estimate the growth inhibition rate caused by the other fungus.
Table 1 “6-82” should be “7-82”?
Indeed, corrected to “7-82”.
Lines 252-255 Please specify which statistical software was used
The following sentence was added to the Materials and Methods section (lines 260-261), as suggested: “Statistics and data analysis were carried out using the JMP program, 7th Edition, SAS Institute Inc., Cary, NC, USA.”
Lines 301-312 there is no need to repeat information from the methodology
The Figure 2 legend was shortened, as advised, to mainly eliminate any unnecessary information that was already presented in the Materials and Methods section and to keep the description brief and scientific.
Lines 334-339 such comments are characteristic of the “Discussion” section, they should not be included in the “Results” section
As suggested by the reviewer, the following phrase was moved and incorporated into the Discussion (lines 481-485): “It is interesting to note that infecting maize plants solely with M. phaseolina caused a 20% reduction in the number of healthy plants compared to the negative control (from 70% to 50%, Table 6). Thus, if both pathogens were acting without interfering with each other, their combined effect (50% in the M. maydis treatment and 20% in the M. phaseolina treatment) should have caused total wilting of the plants. Instead, the dual inoculation led to some recovery of the plants.”
The manuscript contains a lot of figures. Some can be removed, e.g. figs 9, 10
Indeed, Figures 9-10 could be included in the manuscript as supplementary material, as suggested by the reviewer. Nevertheless, the second reviewer did not recommend this and we agree with him. We believe that the disease symptoms provide important information about the disease and the influence of the treatments on the plant’s physiological level, and overall, they strengthen the quantitative results. We prefer leaving this decision to the editor.
Please try to combine tables 6 and 7
Tables 6 and 7 were combined as suggested.